# Individual, community, and structural factors associated with linkage to HIV care among people diagnosed with HIV in Tennessee

Aima A. Ahonkhai[1,2]*, Peter F. Rebeiro[1,3,4], Cathy A. Jenkins[3], Michael Rickles[5], Mekeila Cook[6], Donaldson F. Conserve[7], Leslie J. Pierce[2], Bryan E. Shepherd[3], Meredith Brantley[5], Carolyn Wester[5]

1 Department of Medicine, Infectious Diseases, Vanderbilt University Medical Center, Nashville, TN, United States of America, 2 Vanderbilt Institute for Global Health, Vanderbilt University Medical Center, Nashville, TN, United States of America, 3 Department of Biostatistics, Vanderbilt University School of Medicine, Nashville, TN, United States of America, 4 Department of Medicine, Division of Epidemiology, Vanderbilt University School of Medicine, Nashville, TN, United States of America, 5 Tennessee Department of Health, Nashville, TN, United States of America, 6 Division of Public Health Practice, Meharry Medical College, Nashville, TN, United States of America, 7 Department of Health Promotion, Education and Behavior, University of South Carolina, Columbia, SC, United States of America

* aimalohi.a.ahonkhai@vumc.org

**Data Availability Statement:** Due to the nature of this research where data was collected from the statewide surveillance system, participants of this

## Abstract

### Objective

We assessed trends and identified individual- and county-level factors associated with individual linkage to HIV care in Tennessee (TN).

### Methods

TN residents diagnosed with HIV from 2012–2016 were included in the analysis (n = 3,751). Individuals were assigned county-level factors based on county of residence at the time of diagnosis. Linkage was defined by the first CD4 or HIV RNA test date after HIV diagnosis. We used modified Poisson regression to estimate probability of 30-day linkage to care at the individual-level and the contribution of individual and county-level factors to this outcome.

### Results

Both MSM (aRR 1.23, 95%CI 0.98–1.55) and women who reported heterosexual sex risk factors (aRR 1.39, 95%CI 1.18–1.65) were more likely to link to care within 30-days than heterosexual males. Non-Hispanic Black individuals had poorer linkage than White individuals (aRR 0.77, 95%CI 0.71–0.83). County-level mentally unhealthy days were negatively associated with linkage (aRR 0.63, 95%CI: 0.40–0.99).

### Conclusions

Racial disparities in linkage to care persist at both individual and county levels, even when adjusting for county-level social determinants of health. These findings suggest a need for

study did not agree for their data to be shared publicly, so supporting data is not available. Data can be requested from the Tennessee Department of Health via the following form: https://www.surveygizmo.com/s3/5819792/TDH-Data-Request-Form.

**Funding:** The data in this manuscript have been supported by the National Institute of Allergy and Infectious Diseases (P30-AI110527, AAA) and the National Institute of Mental Health (R25-MH080665, AAA). The funders had no role in study design, data collection and analysis, decision to publish, or preparation of the manuscript.

**Competing interests:** The authors have declared that no competing interests exist.

structural interventions to address both structural racism and mental health needs to improve linkage to care and minimize racial disparities in HIV outcomes.

## Introduction

In 2018, despite representing only 38% of the US population, Southern states accounted for 46% of persons living with HIV (PLWH), and 52% of new diagnoses [1]. To gauge where the gaps in HIV care existed in the southeastern state of Tennessee (TN), the TN Department of Health (TDH) completed its first continuum of care analysis in 2010, revealing that TN under-performed relative to the general US population in both linking newly diagnosed PLWH to care within 90 days (66% vs. 80%) and retaining patients in HIV care over time (37% vs. 46%) [2, 3]. Since then, timely linkage to HIV care has been emphasized by the US Department of Health and Human Services' as a key pillar in the country's Ending the HIV Epidemic (EtE) Initiative, but TN has continued to lag in linkage to care indices [4, 5].

In addition, racial/ethnic disparities have persisted across the US over decades, with non-Hispanic Black (Black) individuals typically experience worse outcomes than other racial/ethnic groups across the entire continuum of HIV care [6]. Racial disparities have been variably attributed to higher rates of poverty, unemployment, and stigma–inequities even more pronounced in the Southern US–and might drive some of TN's poor performance on linkage to HIV care [6–8]. There are limited data characterizing whether and to what extent significant racial disparities in HIV outcomes remain after accounting for both individual and county-level factors known to be associated with poor health outcomes, and disproportionately impacting racial minorities [7, 8]. Such studies are important in the Southern US, home to both a higher rate of incident HIV and more pronounced racial disparities in HIV-related health outcomes than other regions in the US [1, 9].

To improve performance along the HIV care continuum, TDH launched a number of initiatives between 2010 and 2015, including capacity building and infrastructure changes to improve the accuracy and efficiency of HIV testing and reporting, as well as the implementation of a social networking program for Black men who have sex with men (MSM) to address linkage and re-engagement in care [10, 11]. In the wake of these concerted efforts, the objective of this analysis was to integrate individual and county-level data assessing individual, community, and structural drivers of healthcare outcomes to understand 1) trends in linkage to HIV care in TN over time, 2) drivers of poor linkage to care outcomes and 3) drivers of ongoing racial disparities in these outcomes in TN and in counties with the highest HIV burden.

## Materials and methods

### Ethical approval

We obtained a waiver of consent and IRB approval from Vanderbilt University Medical Center (Protocol 173 no. 17119, Nashville, TN, USA), and TDH (protocol no. 1097644–4).

### Study setting and design

We conducted a retrospective cohort study of persons who resided in TN and were newly diagnosed PLWH between January 1, 2012 and December 31, 2016. We assessed trends in linkage to care rates over the study period and individual, community, and structural predictors of linkage to HIV care (measured at the county level). The outcome of interest was linkage

to HIV care, which was defined as receipt of the first CD4 or HIV-1 RNA test result captured via TN's enhanced HIV/AIDS reporting system (eHARS) after the date of diagnosis, and was assessed at 30, 60, and 90 days.

## Measures

**Individual-level measures.** Individual-level variables obtained from eHARS included: year of diagnosis, age at diagnosis (both validated via standardized data cleaning measures to account for repeat testing), sex, race/ethnicity (White/non-Hispanic (White), Black, Hispanic/ all races, other/unknown), HIV risk factor (heterosexual contact, MSM, injection drug use (IDU; includes MSM/IDU), other, unknown), site of diagnosis (inpatient facility/emergency room (ER), outpatient facility, health department or sexually transmitted disease (STD)/family planning clinic, blood bank, correctional facility, other/unknown, missing) and ZIP Code of residence. ZIP Codes in which there were fewer than 5 HIV cases reported were suppressed according to TDH data suppression requirements. We combined sex and HIV risk factor into one variable with the following categories: male/heterosexual, male/MSM, male/IDU, male/ other-unknown, female/heterosexual, female/IDU, and female/other-unknown.

**County-level measures.** Grounded in the social ecological model as a framework to consider barriers to linkage to HIV care, we assessed county-level community and structural factors representing important social determinants of health including healthcare access, socioeconomic status (SES) and disease burden [12, 13]. County was chosen as the unit of measurement because all of the variables of interest were commonly measured or available in aggregate at this level. The measures were drawn from several sources including a CDC-developed Vulnerability Index (VI) which has helped to identify counties at high risk for incident HIV/HCV cases [14, 15]. The VI is comprised of measures such as percent of the population with a car, below the federal poverty level, who are White, have poor or fair health, are smokers, or have a disability. The VI also assesses per capita income, teen birth rate and HIV prevalence. These measures all represent social determinants that may pose barriers to linkage to care for HIV (Table 1) [15]. We retained all 15 variables from the CDC study and included 63 additional collected from the 2010 US Census, as well as TN state-specific indicators from the CDC and TDH surveillance data (Table 2).

Other county-level variables assessed as measures of healthcare access included: percent of the population without health insurance, rate of mental health (MH) providers, per capita urgent care facilities, and per capita primary care physicians [16–18]. MH providers were collected in the 2010 Census and included psychiatrists, psychologists and licensed clinical social workers specializing in MH care. The rate was calculated as the number of MH providers per 100,000 populations. Measures of community socioeconomic status included: percent of the population unemployed, percent of the population with food security, average number of vacant housing units, average number of female-headed households, and average number of drug-related or violent crimes. Finally, measures of community disease burden included: rates of STI diagnosis, percent of HIV cases due to IDU, and average numbers of poor MH days–a measure of community-level mental distress. Average number of mentally unhealthy days was determined using results from the yearly Behavioral Risk Factor Surveillance System (BRFSS) survey that asks participants ". . . thinking about your mental health, . . . how many days during the past 30 days was your mental health not good?" [19]. The BRFSS averages the response to this question at the county-level in accordance with its stratified, probabilistic sampling scheme.

Individuals were assigned exposure status to county-level factors based on county of residence at the time of diagnosis. Additionally, counties with fewer than five PLWH were suppressed by the TDH according to data privacy regulations. We hypothesized that county-level

**Table 1. Demographics of cohort of HIV-positive patients in Tennessee between 2010–2016.**

| Demographic Value | Category | Total | | 30-Day | | 60-Day | | 90-Day | |
|---|---|---|---|---|---|---|---|---|---|
| | | # | % | # | % | # | % | # | % |
| | | (n = 3751) | | (n = 1561) | | (n = 2266) | | (n = 2588) | |
| **Sex** | Male | 2988 | 80% | 1197 | 77% | 1745 | 77% | 2013 | 78% |
| | Female | 763 | 20% | 364 | 23% | 521 | 23% | 575 | 22% |
| **Race/ Ethnicity** | White (Non-Hispanic) | 1231 | 33% | 600 | 38% | 820 | 36% | 935 | 36% |
| | Black (Non-Hispanic) | 2205 | 59% | 801 | 51% | 1223 | 54% | 1408 | 55% |
| | Hispanic (All Races) | 200 | 5% | 98 | 6% | 133 | 6% | 150 | 6% |
| | Other/ Unknown | 115 | 3% | 62 | 4% | 90 | 4% | 95 | 3% |
| **Age at Diagnosis (years)** | Median [IQR] | 31 | [24,43] | 32 | [24,45] | 32 | [24,44] | 32 | [24,44] |
| **HIV Risk Factor** | Heterosexual | 883 | 24% | 378 | 24% | 563 | 25% | 635 | 25% |
| | MSM | 2080 | 55% | 864 | 55% | 1270 | 56% | 1463 | 57% |
| | IDU | 114 | 3% | 54 | 3% | 74 | 3% | 82 | 3% |
| | MSM/IDU | 75 | 2% | 37 | 2% | 49 | 2% | 55 | 2% |
| | Other/ Unknown | 599 | 16% | 228 | 15% | 310 | 14% | 353 | 14% |
| **Year of Diagnosis** | 2012 | 842 | 22% | 333 | 22% | 495 | 22% | 576 | 22% |
| | 2013 | 756 | 20% | 333 | 22% | 488 | 22% | 535 | 21% |
| | 2014 | 729 | 19% | 314 | 20% | 464 | 20% | 529 | 20% |
| | 2015 | 716 | 19% | 305 | 20% | 433 | 19% | 494 | 19% |
| | 2016 | 708 | 19% | 276 | 18% | 386 | 17% | 454 | 18% |
| **Site of Diagnosis** | Inpatient Facility or ER | 746 | 20% | 397 | 25% | 499 | 22% | 746 | 20% |
| | Outpatient Facility | 1291 | 34% | 593 | 38% | 799 | 35% | 1291 | 34% |
| | Health Department or STD/Family Planning Clinic | 1041 | 28% | 349 | 22% | 606 | 27% | 1041 | 28% |
| | Blood Bank | 134 | 4% | 16 | 1% | 39 | 2% | 134 | 4% |
| | Correctional Facility | 195 | 5% | 49 | 3% | 88 | 4% | 195 | 5% |
| | Other/ Unknown | 14 | 0% | 1 | 0% | 4 | 0% | 14 | 0% |
| | Missing | 330 | 9% | 156 | 10% | 231 | 10% | 330 | 9% |

measures of healthcare access (percentage without healthcare insurance, rate of MH providers, per capita urgent care facilities, and per capita primary care physician, percentage of homes with cars), SES (percentage below the federal poverty level, percentage unemployed, average vacant housing units, average number of female-headed households, percentage with food insecurity, violent crime rate), and disease burden (percentage of adults with poor or fair health, percentage of adult smokers, percentage with disability, HIV prevalence, rate of STD diagnosis) would be associated with linkage to HIV care.

## Data analysis

**Individual-level analysis.** Descriptive statistics for demographic and clinical characteristics (median, interquartile range [IQR] or percent, as appropriate) were calculated by linkage to care status within 30, 60 and 90 days of HIV diagnosis, and overall. We used modified Poisson regression to assess risk ratios (RR) for linkage to care at each threshold (30, 60, and 90-days) adjusting for *a priori* selected individual-level covariates in multivariable analysis that are known to be associated with the outcome of interest, including year of- and age at diagnosis, sex, race/ethnicity and HIV risk factor. In the primary analyses, year was modeled as a categorical variable. Age was modeled as a continuous variable using restricted cubic splines with 4 knots to avoid linearity assumptions. Sensitivity analysis was done in which year was modeled as a continuous variable.

**Table 2. Individual-level factors associated with linkage to HIV care.**

| Variable | 30-day | | | 60-day | | | 90-day | | |
|---|---|---|---|---|---|---|---|---|---|
| | aRR* | 95% Confidence Interval | P-Value | aRR* | 95% Confidence Interval | P-Value | aRR* | 95% Confidence Interval | P-Value |
| **Year** | | | 0.34 | | | 0.004 | | | 0.03 |
| 2012 (ref) | 1.00 | | | 1.00 | | | 1.00 | | |
| 2013 | 1.09 | [0.97, 1.22] | | 1.07 | [0.99, 1.16] | | 1.02 | [0.95, 1.08] | |
| 2014 | 1.05 | [0.94, 1.18] | | 1.04 | [0.97, 1.13] | | 1.02 | [0.96, 1.09] | |
| 2015 | 1.08 | [0.96, 1.22] | | 1.03 | [0.95, 1.12] | | 1.01 | [0.94, 1.07] | |
| 2016 | 0.98 | [0.87, 1.11] | | 0.91 | [0.84, 1.00] | | 0.92 | [0.85, 0.99] | |
| **Age at Diagnosis** | | | 0.09 | | | 0.006 | | | <0.001 |
| 20 | 1.09 | [1.01, 1.18] | | 1.06 | [1.00, 1.12] | | 1.07 | [1.02, 1.12] | |
| 25 | 1.01 | [0.98, 1.04] | | 1.00 | [0.98, 1.02] | | 1.00 | [0.98, 1.02] | |
| 30 (ref) | 1.00 | | | 1.00 | | | 1.00 | | |
| 35 | 1.03 | [0.99, 1.07] | | 1.05 | [1.02, 1.08] | | 1.05 | [1.02, 1.07] | |
| 40 | 1.06 | [0.98, 1.15] | | 1.09 | [1.03, 1.15] | | 1.09 | [1.05, 1.14] | |
| 45 | 1.07 | [0.98, 1.17] | | 1.10 | [1.04, 1.17] | | 1.10 | [1.05, 1.16] | |
| **Race/Ethnicity** | | | <0.001 | | | <0.001 | | | <0.001 |
| White, Non-Hispanic (ref) | 1.00 | | | 1.00 | | | 1.00 | | |
| Black, Non-Hispanic | 0.77 | [0.71, 0.83] | | 0.85 | [0.81, 0.90] | | 0.86 | [0.82, 0.90] | |
| Hispanic, All Races | 1.06 | [0.91, 1.23] | | 1.03 | [0.93, 1.15] | | 1.01 | [0.93, 1.10] | |
| Other/Unknown | 1.09 | [0.90, 1.31] | | 1.15 | [1.03, 1.29] | | 1.06 | [0.96, 1.16] | |
| **Sex/Risk Factor** | | | <0.001 | | | <0.001 | | | <0.001 |
| Male/Heterosexual (ref) | 1.00 | | | 1.00 | | | 1.00 | | |
| Male/MSM | 1.15 | [0.99, 1.35] | | 1.09 | [0.98, 1.22] | | 1.05 | [0.96, 1.15] | |
| Male/IDU | 1.23 | [0.98, 1.55] | | 1.11 | [0.94, 1.30] | | 1.05 | [0.93, 1.20] | |
| Male/Other-Unknown | 0.93 | [0.77, 1.13] | | 0.85 | [0.74, 0.97] | | 0.80 | [0.71, 0.89] | |
| Female/Heterosexual | 1.39 | [1.18, 1.65] | | 1.30 | [1.16, 1.46] | | 1.19 | [1.08, 1.30] | |
| Female/IDU | 1.04 | [0.75, 1.45] | | 1.05 | [0.84, 1.31] | | 0.92 | [0.75, 1.13] | |
| Female/Other-Unknown | 1.11 | [0.89, 1.39] | | 1.02 | [0.87, 1.19] | | 1.00 | [0.88, 1.13] | |
| **Site of Diagnosis** | | | <0.001 | | | <0.001 | | | <0.001 |
| Outpatient (ref) | 1.00 | | | 1.00 | | | 1.00 | | |
| Inpatient Facility or ER | 1.18 | 1.08, 1.29] | | 1.09 | [1.02,1.17] | | 1.05 | [0.99, 1.11] | |
| Health Department or STD/Family Planning Clinic | 0.73 | [0.66, 0.81] | | 0.94 | [0.88, 1.01] | | 0.95 | [0.90, 1.01] | |
| Blood Bank | 0.28 | [0.18, 0.44] | | 0.49 | [0.38, 0.64] | | 0.53 | [0.42, 0.66] | |
| Correctional Facility | 0.59 | [0.46, 0.76] | | 0.78 | [0.66, 0.91] | | 0.81 | [0.71, 0.92] | |
| Other/Unknown | 0.18 | [0.03, 1.10] | | 0.52 | [0.23, 1.16] | | 0.56 | [0.28, 1.10] | |
| Missing | 1.00 | [0.88, 1.13] | | 1.10 | [1.01, 1.20] | | 1.07 | [1.00, 1.14] | |

*Adjusted for year of diagnosis, sex/exposure category, race/ethnicity and site of diagnosis.

**County-level analysis.** Individuals were assigned to county-level factors based on county of residence at the time of diagnosis (by merging individual eHARS and county-level data). Modified Poisson regression was used to obtain adjusted RR and marginal probabilities with 95% confidence intervals for the association between county-level characteristics and individual-level linkage outcomes. The models were fit at the individual level, incorporating county-level factors by treating individuals as being nested within counties (and therefore uniformly exposed within counties). We adjusted for individual-level age, sex, race/ethnicity, HIV transmission risk, site of diagnosis and time since HIV diagnosis. These covariates were modeled

using restricted cubic splines for continuous measures and categorical indicators for all other measures. County-level factors were included in multivariable models based on *a priori* identification from CDC's vulnerability index factors, factors associated with healthcare access, and socioeconomic factors as descrived above. We conducted pairwise correlation of all county-level variables and among those that were highly correlated (e.g., correlation < -0.8 or >0.8) only one factor was included to avoid collinearity. Robust standard errors for all models were calculated by clustering at the county level, assuming correlation in the primary outcomes between individuals residing in the same county at the time of HIV diagnosis.

## Results

### Description of cohort of TN residents newly diagnosed PLWH

The data included 3,751 newly diagnosed PLWH in TN between 2012 and 2016. The number of newly diagnosed PLWH gradually decreased from 2012 to 2016 (2012: 842, 2013: 756, 2014: 729, 2015: 716, 2016: 708). Men comprised a greater proportion of the cohort (80%, n = 2988) than women; and Black patients (59%, n = 2205) comprised a greater proportion of the cohort than White (33%, n = 1231) or Hispanic patients (5%, n = 200). The median age at diagnosis was 31 years [IQR 24, 43]. Over half (55%, n = 2080) of the population reported a transmission risk factor of MSM, while 24% (n = 883) reported heterosexual sex and 3% (n = 114) reported IDU. More patients were diagnosed from outpatient facilities (34%, n = 1291) than health department or STD clinics (28%, n = 1041), inpatient facilities or ERs (20%, n = 746), correctional facilities (5%, n = 195) or blood banks (4%, n = 134) (Table 1). Four counties in TN represented 71% of incident cases during the analysis period (Shelby County, county seat of Memphis (n = 1460, 39%); Davidson County, county seat of Nashville (n = 784, 21%); Hamilton County, county seat of Chattanooga (n = 232, 6%); and Knox County, county seat of Knoxville (n = 201, 5%)).

### Trends in establishing HIV care over time

Over the study period, 42% (n = 1,561) of newly diagnosed PLWH were linked to care within 30-, 60% (n = 2,266) within 60-, and 69% (n = 2,588) within 90-days. The proportion of patients linked to care within 30 days of diagnosis increased from 40% (n = 333) in 2012 to 44% (n = 333) in 2013 and decreased to 40% (n = 276) in 2016. Whether linkage to HIV care was defined at 30, 60, or 90 days after HIV diagnosis, linkage increased from 2012 to 2013 then declined to or below the 2012 value by the end of the study period in 2016. As the time to linkage threshold was broadened, the percentage of patients who were linked to an HIV provider increased. Adjusting for other patient-level factors (age, sex, transmission risk factor, site of diagnosis), 30-day linkage to care increased by 13% (aRR 1.13, 95%CI 1.03–1.24), and 60-day linkage to care increased by 9% (aRR 1.09, 95%CI 1.02–1.16) in 2013 compared to 2012. However, the adjusted rate of 30-, 60- and 90-day linkage to care did not significantly differ in 2014, 2015, or 2016 compared to 2012 (with the exception of the risk of 90-day linkage to care in 2016, which decreased by 7% compared to 2012 (aRR 0.93, 95% CI 0.87–0.99)) (Fig 1). When modeled as a linear covariate, year of diagnosis was not a significant predictor of 30-day linkage to care (aRR 0.99, 95% CI 0.97–1.01).

### Individual-level predictors of linkage to care

Age was a good individual predictor of linkage to care. Younger and older patients were more likely to establish care within 30 days (compared to 30 year-olds, aRR 1.09, 1.01, 1.03, 1.06, and 1.07, respectively for ages 20, 25, 35, 40, and 45 years). Race was also an independent predictor

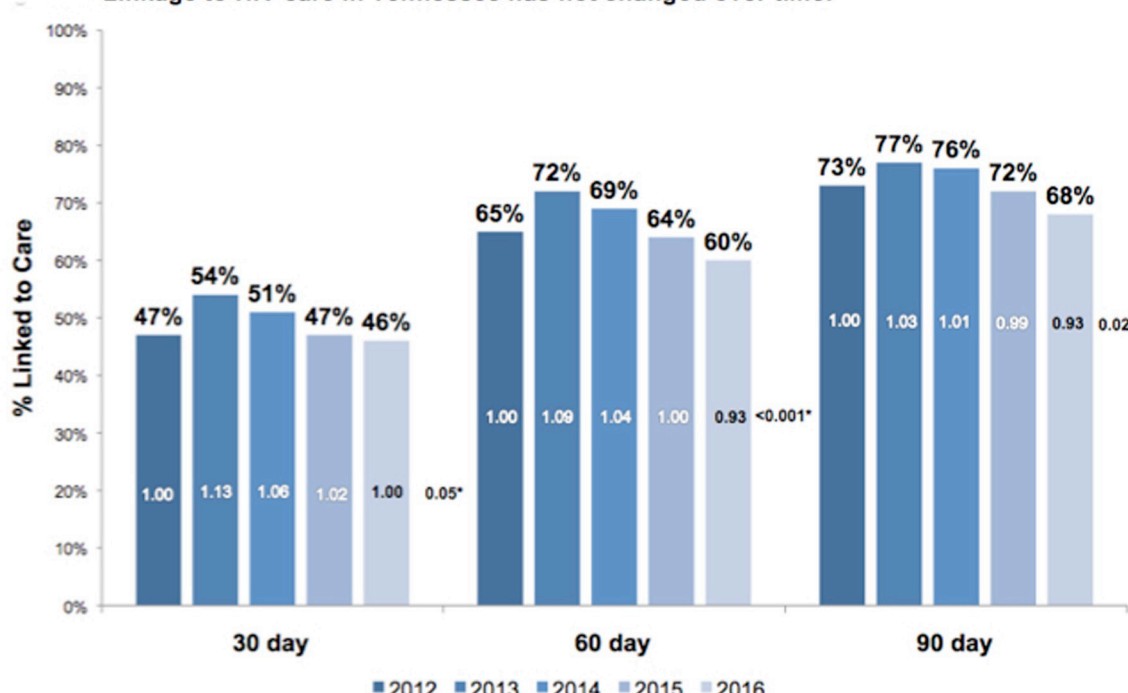

**Fig 1. The rates of 30, 60, and 90-day linkage to HIV Care in Tennessee for patient diagnoses between 2012 and 2016.**

of linkage to care. Black patients had a significantly decreased rate of 30-day linkage to care compared to Whites (aRR 0.77, 95% CI 0.71–0.83). Linkage to care did not differ significantly between White and Hispanic (aRR 1.06, 95%CI 0.91–1.23) or other/unknown patients (aRR 1.09, 95%CI 0.90–1.31). When we combined sex and HIV transmission risk factor categories, we found that heterosexual females (aRR 1.39, 95%CI 1.18–1.65) were more likely to link to HIV care than heterosexual males. Additionally, the location of HIV testing/diagnosis was an important predictor of linkage to care. Compared to an inpatient facility or emergency room, patients diagnosed at inpatient facilities (aRR 1.18, 95%CI 1.08–1.29) were more likely and patients diagnosed at health departments or STD clinics (aRR 0.73, 95%CI 0.66–0.81) and correctional facilities (aRR 0.59, 95%CI 0.46–0.76) were less likely to establish HIV care (Table 2). While the data here are presented for 30-day linkage to care, the same patterns were seen for 60- and 90-day linkage to care as illustrated in Table 2.

## County-level factors associated with linkage to care

Pair wise comparison of association between county-level variables revealed a substantial amount of collinearity. Among the 29 county-level measures assessed, 12 were highly correlated and not included in the model. Accordingly, 17 measures remained in the multivariable model. Only two variables were both clinically and statistically significant in multivariable analysis: Average poor mental health was the strongest county-level predictor of poor linkage care at 30 days (aRR 0.63, 95%CI: 0.40–0.99 per 10-unit increase in poor mental health days). Teen birth rate was also significantly associated with individual linkage to care at 30 days (aRR

1.02, 95%CI: 1.01, 1.04 per 10% increase). For every 10% increase increase in HIV cases due to IDU, individual linkage to care decreased by 4% (aRR 0.91, 95%CI: 0.91–1.00), but this variable did not meet the threshold for statistical significance (Table 3). If one does a Bonferoni adjustment for multiple comparisons, none of the county-level factors remains statistically significant. Notably, White/Non-White segregation index, a variable that reflects greater residential segregation between non-White and White county residents was not included in the final model, but was highly correlated with five of the variables included in final model. Also, in this model which adjusted for both individual and county level factors, White and Hispanic individuals had an increased risk of 30-day linkage to care compared to Black individuals (aRR 1.33, 95%CI 1.30–1.37, aRR 1.44, 95% CI 1.41–1.47 respectively) [data not shown].

## Linkage to care in the highest burden counties in TN

We analyzed the marginal probabilities of linkage at 30-days in the four highest-burden metropolitan counties by race/ethnicity and found that Black patients persistently had the lowest probability of 30-day linkage to care as compared to both White and Hispanic individuals when adjusting for individual level factors, and when adjusting for both individual and county-level factors and when interacting individual-level race/ethnicity with county of residence (Fig 2). Racial disparities were least prominent in Davidson County (the county seat of Nashville), whose residents also had the highest probability of linkage to care of the four highest-HIV-burdened counties in TN.

As in the entire cohort, more patients were diagnosed in outpatient facilities (n = 1025, 38%) than other sites. Thirty-day linkage to care from outpatient facilities was poor across all of 4 counties and ranged from 44% to 53%, and 30-day linkage to HIV care from inpatient or

**Table 3. County level predictors of linking to HIV care within 30 days of diagnosis in Tennessee.**

| Factor | RR [95% CI] |
|---|---|
| Avg. Monthly mental unhealthy days (per 10) | 0.63* [0.40–0.99] |
| Avg. Morphine milligram equivalent (per 1000) | 0.99 [0.98–1.01] |
| Avg. no. drug-related crimes (per 100) | 1.00 [0.99–1.01] |
| Avg. no. drug-related deaths (per 10) | 1.01 [0.94–1.08] |
| Drug trafficking hot-zone | 3.37 [0.88–12.89] |
| No. methadone clinics | 1.06 [0.93–1.20] |
| Per capita income (log10) | 2.95 [0.35–24.78] |
| Per capita primary care physicians (per 10%) | 0.95 [0.74–1.22] |
| Per capita urgent care facilities (per 10%) | 0.51 [0.10–2.60] |
| Percent below FPL | 1.47 [0.07–29.05] |
| Percent of adults smoking (per 10%) | 0.99 [0.86–1.13] |
| Percent of HIV cases due to IDU (per 10%) | 0.96 [0.91–1.00] |
| Percent unemployed | 1.01 [0.96–1.06] |
| Percent with poor/fair health | 1.16 [0.98–1.38] |
| Percent without health insurance | 0.98 [0.94–1.03] |
| Rate mental health providers (per 10%) | 1.05 [0.93–1.20] |
| Teen birth rate (per 10%) | 1.02* [1.01–1.04] |

* p<0.05

*Risk Ratio adjusted for age, sex, race/ethnicity, transmission risk factor, and site of diagnosis.

Avg = Average; No = Number; FPL = Federal poverty line.

IDU = Intravenous drug use; STD = Sexually Transmitted Diseases.

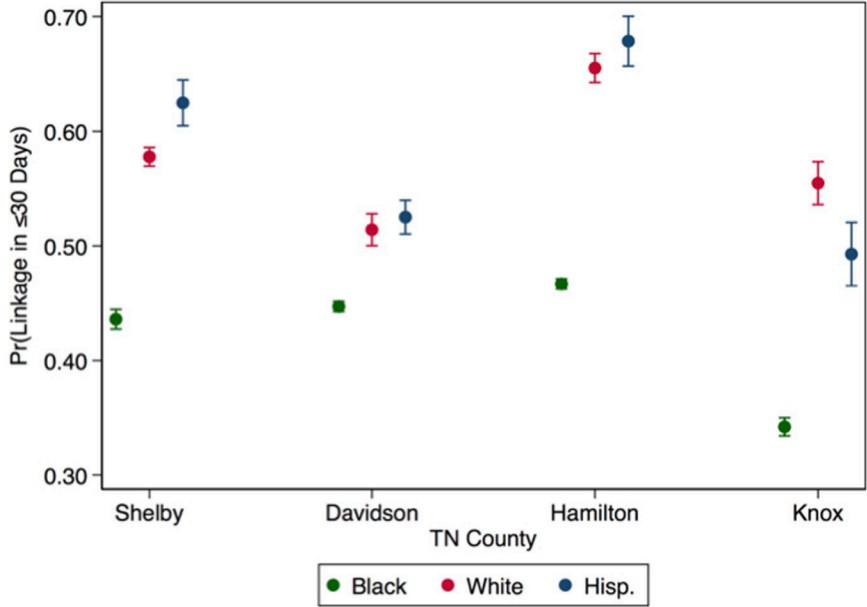

**Panel A: Probability of HIV care linkage within 30 days adjusting for individual-level variables**

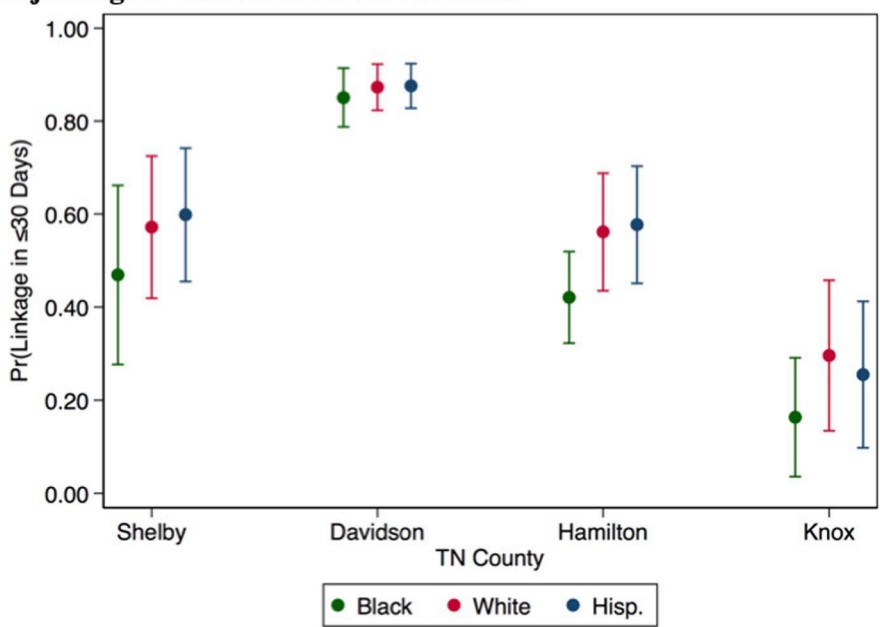

**Panel B: Probability of HIV care linkage within 30 days adjusting for individual *and* county-level variables**

**TN=Tennessee    Hisp=Hispanic**

**Fig 2. The probability of linking to HIV care within 30 days of diagnosis by race/ethnicity for patients living in the four counties with the highest burden of HIV in Tennessee.**

ER facilities ranged from 43% to 59% in each of the four high-burden metropolitan counties showed that the proportion of patients linked to care was more variable and ranged from 43% to 59% (Table 4).

## Discussion

Our analysis of patterns and predictors of linkage to HIV care in TN between 2012 and 2016 highlights unsettling trends. First, despite concerted efforts from TDH, CDC and local partners, timely linkage to HIV care among newly diagnosed PLWH in TN has not only failed to improve over time, but TN now trails the nation in linking PLWH to care. Second, unacceptable racial disparities in linkage to care persist, as Blacks remain much less likely to link to care than Whites–even after accounting for a wide range of individual and structural factors that often are drivers of poor healthcare access and engagement. Our analysis has contributed to the growing call to operationalize measures of structural racism impacting health outcomes by identifying some potential systematic and programmatic opportunities that could be areas for intervention to begin to change this trend in TN [20]. At the same time, our analysis highlights the difficulty of measuring this factor. Indeed, while we assessed 29 county-level variables that all represent social determinants of health, few were statistically significant predictors of individual linkage to care.

In addition to the importance of individual factors like race/ethnicity, our analysis also underscores site of diagnosis as a key predictor of linkage to HIV care. Most studies have highlighted lower linkage to care at testing sites without co-located medical facilities despite higher positivity rates in non-healthcare settings [21–23]. Individuals in our cohort diagnosed at sites without co-located medical facilities, such as correctional facilities and blood banks, were the least likely to link to HIV care. While outpatient facilities yielded the greatest numbers of incident diagnoses in this cohort, inpatients facilities had an 18% increased likelihood of linkage to care. Higher likelihood of linkage to HIV care from inpatient facilities may also reflect the fact that patients diagnosed in these settings are more ill, and thus will more readily establish care after hospitalization. Alternatively, this finding could be artifactual–reflecting routine disease staging with CD4 count and viral load aseessment after diagnosis in the inpatient setting, and not in fact, linkage to care. Nonetheless, linkage to care from both inpatient facilities and outpatient facilities was lowest in Shelby County, the County seat of Memphis, and TN's only priority county nationally targeted for EtE activities. Such findings represent an opportunity for improvement via optimization of linkage referrals and implementation of models such as rapid treatment initiation to promote earlier linkage to care [24].

Our analysis of county-level drivers of linkage to HIV also yielded some intriguing findings. The strongest county-level predictor of linkage to HIV care in TN was the average monthly

**Table 4. County level 30-day linkage to HIV care rates by county and facility type in Tennessee.**

| Facility Type of HIV Diagnosis | Combined | | Shelby | | Knox | | Hamilton | | Davidson | |
|---|---|---|---|---|---|---|---|---|---|---|
| | N | % | N | % | N | % | N | % | N | % |
| Inpatient Facility or Emergency Room | 479 | 48.85% | 276 | 43.48% | 33 | 57.58% | 34 | 58.82% | 135 | 55.14% |
| Outpatient Facility | 1025 | 45.46% | 591 | 44.16% | 53 | 52.83% | 102 | 50.00% | 279 | 45.16% |
| Health Department or STD/Family Planning Clinic | 713 | 33.94% | 315 | 38.10% | 83 | 34.94% | 57 | 45.61% | 258 | 25.97% |
| Blood Bank | 110 | 10.90% | 84 | 9.52% | 4 | 0.00% | 8 | 37.50% | 14 | 7.14% |
| Correctional Facility | 169 | 21.30% | 109 | 18.35% | 24 | 25.00% | 1 | 0.00% | 35 | 28.57% |
| Other/Unknown | 10 | 10.00% | 6 | 0.00% | 1 | 100.00% | 0 | - | 3 | 00.00% |
| Missing | 171 | 47.37% | 79 | 44.30% | 3 | 66.67% | 30 | 53.33% | 59 | 47.46% |

number of mentally unhealthy days. While a rich body of data supports an association between community factors and health outcomes, much of this research has focused on poor SES and other characteristics of neighborhood deprivation [13, 25, 26]. One study set in TN found that individuals living in neighborhoods with the most adverse SES were least likely to achieve virologic suppression [13]. Some authors hypothesize that the relationship between SES and health outcomes are mediated by distribution of stressors, which may be more prevalent in poorer neighborhoods and among racial and ethnic minorities [27, 28]. Others suggest that maladaptive response to stressors may disproportionately impact those with low SES [28]. In our analysis, this association between county mental health and linkage to care was strong, and independent of race.

Surprisingly, access to health insurance at the county level was not significantly associated with linkage to care. While TN has not expanded Medicaid, through the federally-supported Ryan White (RW) program, TN is still able to provide coverage for medical services associated with HIV/AIDS and related illnesses, general insurance assistance and treatment coverage [29]. Across the country, recipients of these funds are more likely to succeed along the continuum of care when compared to uninsured PLWH or those with other forms of healthcare coverage [30, 31]. As such, the promotion of RW services in TN may be an important mechanism to address unmet mental health needs as earlier described. Interestingly, increases in teen birth rate were associated with a small, statistically significant increase in linkage to HIV care. The reasons for this are not entirely clear, but could reflect intense wrap around services for pregnant and peripartum women to prevent mother-to-child HIV transmission or could reflect potential confounding.

It is well-documented that HIV disproportionately affects the Black community in the US at large–a disparity that has persisted decades since the start of the HIV epidemic [6, 8, 9]. Our study findings add to the literature highlighting a critical need to adopt comprehensive strategies to measure drivers of persistent and pervasive racial disparities in HIV outcomes to guide improvement. Today, the life-changing pandemic caused by the novel SARS-coronavirus-2 has targeted a floodlight on the power of structural racism to undermine public health as whole, and to precipitate disparities in COVID-19 infection, hospitalization and death [32, 33]. These trends have furthered important discussions about systemic racial disparities in the US healthcare system and may afford a critical opportunity to seriously consider how to address the structural factors driving such disparities in our healthcare system. In our analysis, the fact that racial disparities persisted despite accounting for both individual characteristics, and as aggregated at the county-level among the four highest-burdened metropolitan areas, speaks to the insidious and complex nature of structural racism. Additionally, the high correlation of residential racial segregation (White, non-White) with many county level factors, while not surprising, further underscores the relationship between race and a range of geographic factors that can impact health. Some, like former president of the American Public Health Association, Dr. Camara Jones, have called on us to recognize structural racism as "a system of structuring opportunity and assigning value based on the social interpretation of how one looks;" and the root cause of all differences in any health outcome associated with race [34]. As such, racism is an important social determinant of health that necessitates a structural intervention [34]. Acknowledging these complex dynamics, several American cities have declared racism as a public health crisis and committed to put racial equity at the core of all city procedures to advocate for policies that improve health in communities of color [35, 36]. Other cities and counties have made similar declarations, but it is clear that they must be accompanied by novel structural approaches to effectively reduce these disparities, and ultimately end the HIV epidemic [37].

Our analysis has notable strengths and weaknesses. Our integration of individual-level surveillance with county-level census and publicly reported data allowed us to identify important individual and county-level risk factors for poor linkage to care, while accounting for many of the socioeconomic drivers of racial disparities in health. However, our use of such data also posed some limitations, as we could not incorporate important factors not readily collected in these systems like individual mental health, experiences with stigma, racism, SES and other barriers to healthcare access and earlier linkage to HIV care for PLWH in TN. We were also unable to distinguish transgender individuals who are at great risk for poor health outcomes. Additionally, despite the improvements in HIV surveillance and data quality since 2012, our measures of linkage to care were reliant on the completeness of the mandatory reporting system, which varies by site and could have introduced some bias. We acknowledge that we have included many covariates in our analyses, and could be subject to limitations from multiple testing. Finally, more granular spatial analysis (i.e. ZIP Code rather than county-level) was limited by both data suppression requirements for TDH and a lack of available public health data at the zip-code level.

## Conclusions

In conclusion, to meet the critical EtE target of reducing the number of new HIV infections in the US by 90% and move towards ending the epidemic, statewide linkage to care in TN needs to improve. Despite targeted efforts both broadly and in minority communities, linkage to HIV care did not improve substantially from 2012 to 2016. Racial disparities that persist at both individual and county levels suggest the need for exploring structural interventions to address racism as a public health threat. In addition, optimizing outreach for young heterosexual men who may be overlooked by interventions targeting MSM, and addressing linkage to care processes from outpatient and community-based testing facilities through improved partnerships or co-location of testing and treatment services are potential areas for intervention. Further exploration of the role of poor community and individual mental health in this environment is needed to inform mental health interventions to improve engagement in HIV care.

## Author Contributions

**Conceptualization:** Aima A. Ahonkhai.

**Formal analysis:** Peter F. Rebeiro, Cathy A. Jenkins, Bryan E. Shepherd.

**Funding acquisition:** Aima A. Ahonkhai.

**Investigation:** Aima A. Ahonkhai, Michael Rickles, Mekeila Cook, Donaldson F. Conserve, Meredith Brantley, Carolyn Wester.

**Methodology:** Aima A. Ahonkhai, Peter F. Rebeiro, Cathy A. Jenkins.

**Project administration:** Leslie J. Pierce.

**Supervision:** Aima A. Ahonkhai, Bryan E. Shepherd, Carolyn Wester.

**Validation:** Aima A. Ahonkhai.

**Visualization:** Aima A. Ahonkhai, Leslie J. Pierce.

**Writing – original draft:** Aima A. Ahonkhai, Leslie J. Pierce.

**Writing – review & editing:** Aima A. Ahonkhai, Peter F. Rebeiro, Cathy A. Jenkins, Michael Rickles, Mekeila Cook, Donaldson F. Conserve, Leslie J. Pierce, Bryan E. Shepherd, Meredith Brantley, Carolyn Wester.

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
