## [Decision Letter · Decision Letter 0]

22 Jul 2021

PONE-D-21-13112

Individual, Community, and Structural Factors Associated with Linkage to HIV Care Among People Diagnosed with HIV in Tennessee

PLOS ONE

Dear Dr. Ahonkhai,

Thank you for submitting your manuscript to PLOS ONE. After careful consideration, we feel that it has merit but does not fully meet PLOS ONE’s publication criteria as it currently stands. Therefore, we invite you to submit a revised version of the manuscript that addresses the points raised during the review process. In your response, please pay particular attention to the concerns regarding the definition of linkage being used in the paper, as well as clarification of the modeling approach and interpretation of the study findings in light of structural factors.

We look forward to receiving your revised manuscript.

Kind regards,

Becky L. Genberg

Academic Editor

PLOS ONE

Journal Requirements:

2. In your Methods section, please ensure that the data sources and codes used are described in adequate detail.

Reviewers' comments:

Reviewer's Responses to Questions

**Comments to the Author**

1. Is the manuscript technically sound, and do the data support the conclusions?

Reviewer #1: Partly

Reviewer #2: Partly

2. Has the statistical analysis been performed appropriately and rigorously? 

Reviewer #1: I Don't Know

Reviewer #2: Yes

3. Have the authors made all data underlying the findings in their manuscript fully available?

Reviewer #1: No

Reviewer #2: No

4. Is the manuscript presented in an intelligible fashion and written in standard English?

Reviewer #1: Yes

Reviewer #2: Yes

5. Review Comments to the Author

Reviewer #1: Overview:

This study evaluates individual and county-level predictors of linkage to HIV care following new HIV diagnosis in Tennessee over time. The manuscript addresses an important topic and adds particular value by identifying structural drivers of poor linkage and persistent racial disparities using county-level data. The paper is well written and I think makes an important contribution, though I have some concerns about the definition of linkage to care and its interpretation. My comments/suggested edits are below:

Major:

1. The authors define linkage as the date that the first CD4 count or HIV viral load measurement is taken including those drawn on the same date that the HIV diagnosis was made. While this certainly could represent linkage to HIV care, particularly if the diagnosis was made in an outpatient facility that also provides chronic HIV care, it equally could represent labs that are drawn in a non-HIV care facility (e.g. emergency department or inpatient hospital) at the time the diagnosis was made. Given that this likely does not constitute true linkage, I think the authors should consider incorporating sensitivity analyses where CD4 count/viral load measures within 24 hours of diagnosis from inpatient/emergency department or other episodic/urgent care settings (e.g. blood banks) are excluded from the linkage definition. Additional more granular information about the time to linkage following diagnosis would also be helpful – how commonly was the definition of linkage met on the same day as the diagnosis and how did this differ by care setting?

2. Given the above uncertainties with the linkage definition, I suggest incorporating a second outcome of the proportion achieving viral suppression by 6 and/or 12 months after diagnosis. This measures a different process than linkage alone, however it would strengthen study findings, particularly if disparities are similar between those who are less likely to link and those less likely to attain viral suppression.

Minor:

1. Line 46 – consider changing ‘performing’ to something like ‘experience worse outcomes’. The wording here implies that non-Hispanic Black individuals are to blame for ‘underperforming’ in the care cascade, whereas the authors clearly intend to shed light on racial disparities and ways that the healthcare system is underperforming for these individuals.

2. Line 54 – “US” is duplicated, remove one instance.

3. Some of the county-level factors appear to have very extreme point estimates (e.g. ‘percent experiencing food insecurity’ with RR of 4.7 million and ‘percent households with a car with RR of 0.00). Can the authors double check these results and provide some explanation for the extreme values here?

Reviewer #2: The authors assessed trends and factors associated with linkage to care among TN residents diagnosed with HIV from 2012 to 2016. The article was well-written. And the incorporation of a vast number of community-level exposures is a nice addition to the literature. My comments are detailed below.

Abstract. The methods and discussion in the abstract suggest you ran county-level models. However, based on the methods in the narrative, it seems you ran a county-level analysis using individual data (i.e., using individual data with county-level independent variables). Is this an ecologic analysis?

Methods: In general, I think the modelling approach requires clarification. And the approach for incorporating the county-level variables requires clarification.

1. Study, setting and design. How were county-level measures assigned or merged with eHARS data? Is the variable merged to an individual based on county of diagnosis or county of residence? Is this ecologic data: do you have measure county-level linkage and merge with a county-level exposure dataset?

2. Measures. For a variable like poor mental health days, did you use the average for the county? How variable is the measure – would the median be a better metric for a county?

3. Measures. Are all of the community SES variables a percent in the county? And were you able to drill down further, say to the zipcode?

4. Individual-level analysis. What are the a priori covariates in the multivariable analysis? And how/why were they chosen? Based on the tables, I think you fit three models, one for each threshold, with the variables listed under table 2, but this could be clearer in the narrative.

5. County-level analysis. You have listed a lot of independent variables (N = 23 county level). Did you fit a model for each independent variable/exposure of interest? If not, how did you address collinearity? Current approach suggests a need to correct for a lot of multiple testing – how was this addressed?

6. County-level analysis. How were model covariates selected? And was the approach for selecting model covariates different than the approach for selecting model covariates in the individual-level analysis. If so, why?

Discussion. My primary concern with this discussion is that you’ve zeroed in on the significant findings and given very little consideration to your mostly null county-level associations. Usually, I think that’s fine, but in this case, I worry a reader may consider it fishing for a county-level association. Is it possible that the county-level variables are measured at too wide of a geographic level, e.g., would a zip-code level exposure be better?

1. The inpatient finding seems artifactual, given persons diagnosed while at an inpatient facility are admitted / on-site. I might re-frame this to look at outpatient as your reference. It seems more actionable to know if the health department or correctional facilities perform as well as outpatient facilities (table 2 suggests they perform worse).

2. Why declining linkage by 2016? What was happening or stopped happening – any TN DOH initiatives?

3. Minor / editorial, but I would suggest using a word other than “incited” to highlight the increase in discussions about racial disparities (discussion, paragraph 2). Is it possible to cite HIV literature on structural determinants versus COVID?

4. The structural racism angle/paragraph requires additional work. Your paper is about individual and community exposures. How do you tie them to structural racism (as a root cause)? And largely, your community level measures had a null association with linkage. Does this support your theory of structural racism as a root cause? Do you consider these county measures proxies for structural racism?

5. Limitations. I would add multiple testing. Are county level variables too broad of an exposure?

6. PLOS authors have the option to publish the peer review history of their article (what does this mean?). If published, this will include your full peer review and any attached files.

Reviewer #1: No

Reviewer #2: No

---

## [Author Response · Author response to Decision Letter 0]

29 Nov 2021

Editor Comments

We have edited the manuscript to ensure that it meets PLOS ONE’s style requirements. 

2. In your Methods section, please ensure that the data sources and codes used are described in adequate detail.

We have updated the methods section to ensure that the data sources and codes are clearly described. 

3. In your Data Availability statement, you have not specified where the minimal data set underlying the results described in your manuscript can be found. PLOS defines a study's minimal data set as the underlying data used to reach the conclusions drawn in the manuscript and any additional data required to replicate the reported study findings in their entirety. All PLOS journals require that the minimal data set be made fully available. For more information about our data policy, please see http://journals.plos.org/plosone/s/data-availability. Upon re-submitting your revised manuscript, please upload your study’s minimal underlying data set as either Supporting Information files or to a stable, public repository and include the relevant URLs, DOIs, or accession numbers within your revised cover letter. For a list of acceptable repositories, please see http://journals.plos.org/plosone/s/data-availability#loc-recommended-repositories. Any potentially identifying patient information must be fully anonymized.

These data cannot be shared publicly according to CDC HIV/STD policy and Tennessee law.

4. Important: If there are ethical or legal restrictions to sharing your data publicly, please explain these restrictions in detail. Please see our guidelines for more information on what we consider unacceptable restrictions to publicly sharing data: http://journals.plos.org/plosone/s/data-availability#loc-unacceptable-data-access-restrictions. Note that it is not acceptable for the authors to be the sole named individuals responsible for ensuring data access. We will update your Data Availability statement to reflect the information you provide in your cover letter.

CDC HIV/STD security and confidentiality policy and Tennessee law prohibits the public release of de-identified, client-level HIV surveillance data. Publicly available information on persons newly diagnosed with HIV can be found here: https://www.tn.gov/health/health-program-areas/statistics/health-data/hiv-data.html. We appreciate you updating this information in our Data Availability statement.

Reviewer 1 Comments

1. The authors define linkage as the date that the first CD4 count or HIV viral load measurement is taken including those drawn on the same date that the HIV diagnosis was made. While this certainly could represent linkage to HIV care, particularly if the diagnosis was made in an outpatient facility that also provides chronic HIV care, it equally could represent labs that are drawn in a non-HIV care facility (e.g. emergency department or inpatient hospital) at the time the diagnosis was made. Given that this likely does not constitute true linkage, I think the authors should consider incorporating sensitivity analyses where CD4 count/viral load measures within 24 hours of diagnosis from inpatient/emergency department or other episodic/urgent care settings (e.g. blood banks) are excluded from the linkage definition. Additional more granular information about the time to linkage following diagnosis would also be helpful – how commonly was the definition of linkage met on the same day as the diagnosis and how did this differ by care setting?

We agree with the challenges posed with our initial diagnosis of linkage, and have updated the definition. Linkage is now defined at the date of the first CD4 count or viral load after HIV diagnosis. The methods have been updated (Page 3, Lines 72-74) to reflect this. The results have not substantially changed (neither effect size nor direction of responses). 

The outcome of interest was linkage to HIV care, which was defined as receipt of the first CD4 or HIV-1 RNA test result captured via TN’s enhanced HIV/AIDS reporting system (eHARS) after diagnosis, and was assessed at 30, 60, and 90 days. 

2. Given the above uncertainties with the linkage definition, I suggest incorporating a second outcome of the proportion achieving viral suppression by 6 and/or 12 months after diagnosis. This measures a different process than linkage alone, however it would strengthen study findings, particularly if disparities are similar between those who are less likely to link and those less likely to attain viral suppression.

As described in the response to Reviewer 1 Comment 1, we have updated the definition of linkage to care. Linkage is now defined at the date of the first CD4 count or viral load after HIV diagnosis. The methods have been updated (Page 3, Lines 72-74) to reflect this. The results have not substantially changed (neither effect size nor direction of responses).

3. Line 46 – consider changing ‘performing’ to something like ‘experience worse outcomes’. The wording here implies that non-Hispanic Black individuals are to blame for ‘underperforming’ in the care cascade, whereas the authors clearly intend to shed light on racial disparities and ways that the healthcare system is underperforming for these individuals.

We have edited this text as suggested (Page 2, Line 47) which now reads as below:

In addition, racial/ethnic disparities have persisted across the US over decades, with non-Hispanic Black (Black) individuals typically experience worse outcomes than other racial/ethnic groups across the entire continuum of HIV care.

4. Line 54 – “US” is duplicated, remove one instance.

We have fixed this typographical error. 

5. Some of the county-level factors appear to have very extreme point estimates (e.g. ‘percent experiencing food insecurity’ with RR of 4.7 million and ‘percent households with a car with RR of 0.00). Can the authors double check these results and provide some explanation for the extreme values here?

These extreme results were due to collinearity in our analyses (see response to Reviewer 2, comment 6). We have updated the analysis to account for collinearity among county-level factors, by excluding some county level factors that are highly correlated with each other. Withthis approach, estimates are more stable. Also, percent households with a car is no longer included in the model. 

Reviewer 2 Comments: 

1. Abstract. The methods and discussion in the abstract suggest you ran county-level models. However, based on the methods in the narrative, it seems you ran a county-level analysis using individual data (i.e., using individual data with county-level independent variables). Is this an ecologic analysis?

We appreciate the opportunity to clarify this point. This is not an ecological analysis, as we assessed individual-level outcomes. We have updated the methods section in the abstract (Page 1, Lines 22-25) to clarify the modeling/analysis approach.

TN residents diagnosed with HIV from 2012-2016 were included in the analysis (n=3,751). Individuals were assigned to county-level factors based on county of residence at the time of diagnosis. Linkage was defined by the first CD4 or HIV RNA test date after HIV diagnosis. We used modified Poisson regression to estimate probability of 30-day linkage to care at the individual-level, and the contribution of individual and county-level factors to this outcome.

2. Methods: In general, I think the modelling approach requires clarification. And the approach for incorporating the county-level variables requires clarification. Study, setting and design. How were county-level measures assigned or merged with eHARS data? Is the variable merged to an individual based on county of diagnosis or county of residence? Is this ecologic data: do you have measure county-level linkage and merge with a county-level exposure dataset?

We have updated the methods (Page 5, Lines 144-158) to clarify the modeling approach, and how individual and county-level measures were merged. This section now reads as below:

Individuals were assigned to county-level factors based on county of residence at the time of diagnosis (by merging individual eHARS and county-level data). Modified Poisson regression was used to obtain adjusted RR and marginal probabilities with 95% confidence intervals for the association between county-level characteristics and individual-level linkage outcomes. The models were fit at the individual level, incorporating county-level factors by treating individuals as being nested within counties (and therefore uniformly exposed within counties). We adjusted for individual-level age, sex, race/ethnicity, HIV transmission risk, site of diagnosis and time since HIV diagnosis. These covariates were modeled using restricted cubic splines for continuous measures and categorical indicators for all other measures. County-level factors were included in multivariable models based on a priori identification from CDC’s vulnerability index factors, factors associated with healthcare access, and socioeconomic factors as descrived above. Models were fit including all county-level variables from each of these domains separately, and then together across all county-level domains in a full multivariable model. Robust standard errors for all models were calculated by clustering at the county level, assuming correlation in the primary outcomes between individuals residing in the same county at the time of HIV diagnosis. 

3. Measures. For a variable like poor mental health days, did you use the average for the county? How variable is the measure – would the median be a better metric for a county?

The Behavioral Risk Factor Surveillance Survey provides average results at the county level (as well as results averaged across other demographic factors). It is not possible to extract different measures of central tendency at this stage, as this county-level averaging is done centrally after applying the appropriate weights to each surveillance sample. We have updated the methods section (Page 4, Lines 116-118) to reflect this as below:

Average number of mentally unhealthy days was determined using results from the yearly Behavioral Risk Factor Surveillance System (BRFSS) survey that asks participants “… thinking about your mental health, … how many days during the past 30 days was your mental health not good?”19 The BRFSS averages the response to this question at the county-level, in accordance with its stratified, probabilistic sampling scheme. 

4. Measures. Are all of the community SES variables a percent in the county? And were you able to drill down further, say to the zipcode?

County was chosen as the unit of geographic analysis for two reasons. First, some of the geographic measures were available at the county but not ZIP-Code-level. Second, due to data privacy policies, Tennessee Department of Health could not share patient-level data for regions with fewer than 5 HIV cases, hence using a smaller geographic area would have created more missing data in the analysis. We have updated the methods (Pages 3 and 4, Lines 93-95 and 120-122), and descrbied this as a limitation in the Discussion (Page 15, Lines 345-347). 

Methods

Grounded in the social ecological model as a framework to consider barriers to linkage to HIV care, we assessed county-level community and structural factors representing healthcare access, socioeconomic status (SES) and disease burden.12,13 County was chosen as the unit of measurement because all variables of interest were commonly measured or available in aggregate at this level. The measures were drawn from several sources including a CDC-developed Vulnerability Index (VI) which has helped to identify counties at high risk for incident HIV/HCV cases.14,15 The VI is comprised of measures such as percent of the population with a car, below the federal poverty level, who are White, have poor or fair health, are smokers, or have a disability. The VI also assesses per capita income, teen birth rate and HIV prevalence. These measures all represent social determinants that may pose barriers to linkage to care for HIV [Table 1].15 We retained all 15 variables from the CDC study and included 63 additional collected from the 2010 US Census, as well as TN state-specific indicators from the CDC and TDH surveillance data [Table 2].

Individuals were assigned exposure status to county-level factors based on county of residence. Additionally, counties with fewer than five PLWH were suppressed by the TDH according to data privacy regulations.

Discussion

Finally, more granular spatial analysis (i.e., ZIP Code rather than county-level) was limited by both data suppression requirements for TDH, and lack of available public health data at the zip-code level.

5. Individual-level analysis. What are the a priori covariates in the multivariable analysis? And how/why were they chosen? Based on the tables, I think you fit three models, one for each threshold, with the variables listed under table 2, but this could be clearer in the narrative.

Individual-level covariates were limited to available surveillance data collected. These demographic factors (age, sex, race-ethnicity, transmission risk factor) are also often associated with HIV care outcomes, so all were included in the analysis. We have updated the methods section (Page 5, Lines 135-141) to better explain the analysis plan.

We used modified Poisson regression to assess risk ratios (RR) for linkage to care at each threshold (30, 60, and 90-days) adjusting for a priori selected individual-level covariates in multivariable analysis that were available in the surveillance data and known to be associated with the outcome of interest, including year of and age at diagnosis, sex, race/ethnicity and HIV transmission risk factor.

6. County-level analysis. You have listed a lot of independent variables (N = 23 county level). Did you fit a model for each independent variable/exposure of interest? If not, how did you address collinearity? 

Our previous analyses did not account for collinearity, which was an oversight by us and has been fixed in the revised manuscript. Many of these county-level variables are highly correlated. In our revised analysis, we looked at the pairwise correlation of all county-level variables and among those that were highly correlated (e.g., correlation < -0.8 or >0.8) we selected one county-level variable to include in the model; our choice was based on our perceived scientific interest. The Methods (Page 5, Lines 154-158) and Results (Page 11, Lines 217-229) sections have been updated accordingly. 

Methods

We conducted pairwise correlation of all county-level variables and among those that were highly correlated (e.g., correlation < -0.8 or >0.8) only one factor was included to avoid collinearity.

 Results

Pair wise comparison of association between county-level variables revealed a substantial amount of collinearity. Among the 29 county-level measures assessed, 12 were highly correlated and not included in the model. Accordingly, 17 measures remained in the multivariable model. Only two variables were both clinically and statistically significant in multivariable analysis: Average poor mental health was the strongest county-level predictor of poor linkage care at 30 days (aRR 0.63, 95%CI: 0.40-0.99 per 10-unit increase in poor mental health days). Teen birth rate was also significantly associated with individual linkage to care at 30 days (aRR 1.02, 95%CI: 1.01, 1.04 per 10% increase). For every 10% increase increase in HIV cases due to IDU, individual linkage to care decreased by 4% (aRR 0.91, 95%CI: 0.91-1.00), but this variable did not meet the threshold for statistical significance [Table 3]. If one does a Bonferoni adjustment for multiple comparisons, none of the county-level factors remains statistically significant. Notably, White/Non-White segregation index, a variable that reflects greater residental segregation between non-White and White county residents was not included in the final model, but was highly correlated with five of the variables included in final model [data not included].

Table 3: County level predictors of linking to HIV care within 30 days of diagnosis in Tennessee

Factor RR [95% CI]

Avg. Monthly mental unhealthy days (per 10) 0.63* [0.40-0.99]

Avg. Morphine milligram equivalent (per 1000) 0.99 [0.98-1.01]

Avg. no. drug-related crimes (per 100) 1.00 [0.99-1.01]

Avg. no. drug-related deaths (per 10) 1.01 [0.94-1.08]

Drug trafficking hot-zone 3.37 [0.88-12.89]

No. methadone clinics 1.06 [0.93-1.20]

Per capita income (log10) 2.95 [0.35-24.78]

Per capita primary care physicians (per 10%) 0.95 [0.74-1.22]

Per capita urgent care facilities (per 10%) 0.51 [0.10-2.60]

Percent below FPL 1.47 [0.07-29.05]

Percent of adults smoking (per 10%) 0.99 [0.86-1.13]

Percent of HIV cases due to IDU (per 10%) 0.96 [0.91-1.00]

Percent unemployed 1.01 [0.96-1.06]

Percent with poor/fair health 1.16 [0.98-1.38]

Percent without health insurance 0.98 [0.94-1.03]

Rate mental health providers (per 10%) 1.05 [0.93-1.20]

Teen birth rate (per 10%) 1.02* [1.01-1.04]

* p<0.05

7. Current approach suggests a need to correct for a lot of multiple testing – how was this addressed?

We acknowledge that we have included many covariates in our analyses, and that there may be an issue with multiple testing. In general, we usually do not like to adjust for multiple comparisions because it is not straightforward to determine exactly how many comparisons were performed (does one count per outcome [30-day, 60-day, or 90-day linkage], or per table, or over the entire manuscript). Our preference is to present all results and then to let reviewers decide how to interpret them. However, we agree that it is worth pointing out the multiple comparisons to readers. In the revision, we added a sentence in the Results when discussing the county-level factors (Page 11, Lines 225-227): 

If one does a Bonferoni adjustment for multiple comparisons, none of the factors remains statistically significant.

8. County-level analysis. How were model covariates selected? And was the approach for selecting model covariates different than the approach for selecting model covariates in the individual-level analysis. If so, why?

County-level variables were purposefully selected to reflect healthcare access, community socioeconomic status, and community disease burden. We also included all 15 variables in the CDC-developed Vulnerability Index (VI) which has helped to identify counties at high risk for incident HIV/HCV cases. We hypothesized that these variables would be associated not only with new HIV infection, but also with linkage toHIV care. All variables included reflect important social determinants of health. We have edited the methods (Page 3, Lines 91-93) to clarify this.

Grounded in the social ecological model as a framework to consider barriers to linkage to HIV care, we assessed county-level community and structural factors representing important social determinants of health including healthcare access, socioeconomic status (SES) and disease burden.

9. Discussion. My primary concern with this discussion is that you’ve zeroed in on the significant findings and given very little consideration to your mostly null county-level associations. Usually, I think that’s fine, but in this case, I worry a reader may consider it fishing for a county-level association. Is it possible that the county-level variables are measured at too wide of a geographic level, e.g., would a zip-code level exposure be better?

We recognize the reviewer’s concern, and have updated the methods to make it clearer that the county-level measures chosen were purposefully selected to represent important social determinants of health. We also explain obstacles to conducting a ZIP-Code-level analysis (Reviewer 2 Comments 4 and 8). Both of these responses are repeated below for ease of reference.

County was chosen as the unit of geographic analysis for two reasons. First, some of the geographic measures were available at the county but not ZIP-Code-level. Second, due to data privacy policies, Tennessee Department of Health could not share patient-level data for regions with fewer than 5 HIV cases, hence using a smaller geographic area would have created more missing data in the analysis. We have updated the methods (Page 3, Lines 94-95), and described this as a limitation in the Discussion (Page 15, Lines 345-347).

Methods

Grounded in the social ecological model as a framework to consider barriers to linkage to HIV care, we assessed county-level community and structural factors representing healthcare access, socioeconomic status (SES) and disease burden.12,13 County was chosen as the unit of measurement because all of the variables of interest were measured or aggregated at this level. The measures were drawn from several sources including a CDC-developed Vulnerability Index (VI) which has helped to identify counties at high risk for incident HIV/HCV cases.14,15 The VI is comprised of measures such as percent of the population with a car, below the federal poverty level, who are White, have poor or fair health, are smokers, or have a disability. The VI also assesses per capita income, teen birth rate and HIV prevalence. These measures all represent social determinants that may pose barriers to linkage to care for HIV [Table 1].15 We retained all 15 variables from the CDC study and included 63 additional collected from the 2010 US Census, as well as TN state-specific indicators from the CDC and TDH surveillance data [Table 2]. Individuals were assigned exposure status to county-level factors based on county of residence. Additionally, counties with fewer than five PLWH were suppressed by the TDH according to data privacy regulations.

Discussion

Finally, more granular spatial analysis (ie zip code rather than county-level) was limited by both data suppression requirements for TDH, and lack of available public health data at the zip-code level.

County-level variables were purposefully selected to reflect healthcare access, community socioeconomic status, and community disease burden. We also included all 15 variables in the CDC-developed Vulnerability Index (VI) which has helped to identify counties at high risk for incident HIV/HCV cases. We hypothesized that these variables would be associated not only with new HIV infection, but also with linkage toHIV care. All variables included reflect important social determinants of health. We have edited the methods (Page 3, Lines 94-95) to clarify this.

Grounded in the social ecological model as a framework to consider barriers to linkage to HIV care, we assessed county-level community and structural factors representing important social determinants of health including healthcare access, socioeconomic status (SES) and disease burden.

10. The inpatient finding seems artifactual, given persons diagnosed while at an inpatient facility are admitted / on-site. I might re-frame this to look at outpatient as your reference. It seems more actionable to know if the health department or correctional facilities perform as well as outpatient facilities (table 2 suggests they perform worse).

We have updated the analysis to make the outpatient facility the reference group. The updated findings are presented in the Results section (Page 8, Lines 207-210), and Table 2.We agree that this finding could be artifactual depending on the reason for admission, and hospital practices. We have included this rationale in the discussion (Page 13, Lines 279-283). 

Results

Compared to an outpatient facility or emergency room, patients diagnosed at inpatient facilities (aRR 1.18, 95%CI 1.08-1.29) were more likely and patients diagnosed at health departments or STD clinics (aRR 0.73, 95%CI 0.66-0.81) and correctional facilities (aRR 0.59, 95%CI 0.46-0.76) were less likely to establish HIV care [Table 2]. 

 Discussion

Higher likelihood of linkage to HIV care from inpatient facilities may also reflect the fact that patients diagnosed in these settings are more ill, and thus will more readily establish care after hospitalization. Higher linkage to care from inpatient facilities could alternatively be artifactual – reflecting routine disease staging with CD4 count and viral load aseessment after diagnosis, and not in fact, linkage to care. Nonetheless, linkage to care from both inpatient facilities and outpatient facilities was lowest in Shelby County, the County seat of Memphis, and TN’s only priority county nationally targeted for EtE activities. Such findings represent an opportunity for improvement via optimization of linkage referrals at such sites, and implementation of models such as rapid treatment initiation to promote earlier linkage to care.27 

11. Why declining linkage by 2016? What was happening or stopped happening – any TN DOH initiatives?

We have reviewed these data with our partners at TDH. The reasons for declining linkage in 2016 are likely multifactorial and could have included changes in lab reporting and surveillance practices especially with testing partners. As this was beyond the scope of our analysis, we have not opined on this specifically in the discussion. 

12. Minor / editorial, but I would suggest using a word other than “incited” to highlight the increase in discussions about racial disparities (discussion, paragraph 2). Is it possible to cite HIV literature on structural determinants versus COVID?

We have changed this word as suggested in the Discussion (Page 15, Line 318).

These trends have furthered important discussions about systemic racial disparities in the US healthcare system, and may afford a critical opportunity to seriously consider how to address the structural factors driving such disparities in our healthcare system. 

13. The structural racism angle/paragraph requires additional work. Your paper is about individual and community exposures. How do you tie them to structural racism (as a root cause)? And largely, your community level measures had a null association with linkage. Does this support your theory of structural racism as a root cause? Do you consider these county measures proxies for structural racism?

We have reframed this section and reorganized the discussion to better address these concerns. Our paper was about individual and community exposures, but the county-level exposures do represent important social determinants of health which are accepted drivers of structural racism. Our manuscript does add to necessary efforts to operationalize measures of structural racism – a complex challenge. The discussion was updated to reflex this (Page 13, Lines 266-268, and Page 15, Lines 313-334). 

Our analysis of patterns and predictors of linkage to HIV care in TN between 2012 and 2016 highlights unsettling trends. First, despite concerted efforts from TDH, CDC and local partners, timely linkage to HIV care among newly diagnosed PLWH in TN has not only failed to improve over time, but TN now trails the nation in linking PLWH to care. Second, unacceptable racial disparities in linkage to care persist, as Blacks remain much less likely to link to care than Whites – even after accounting for a wide range of individual and structural factors that often are drivers of poor healthcare access and engagement. Our analysis has contributed to the growing call to operationalize measures of structural racism impacting health outcomes by identifying some potential systematic and programmatic opportunities that could be areas for intervention to begin to change this trend in TN.20 At the same time, our analysis highlights the difficulty of measuring this factor. Indeed, while we assessed 29 county-level variables that all represent social determinants of health, few were statistically significant predicotrs of individual linkage to care. 

It is well-documented that HIV disproportionately affects the Black community in the US at large – a disparity that has persisted decades since the start of the HIV epidemic.6,8,9 Our study findings add to the literature highlighting a critical need to adopt comprehensive strategies to measure drivers of persistent and pervasive racial disparities in HIV outcomes to guide improvement. Today, the life-changing pandemic caused by the novel SARS-coronavirus-2 has targeted a floodlight on the power of structural racism to undermine public health as whole, and to precipitate disparities in COVID-19 infection, hospitalization, and death.35,36 These trends have furthered important discussions about systemic racial disparities in the US healthcare system, and may afford a critical opportunity to seriously consider how to address the structural factors driving such disparities in our healthcare system. In our analysis, the fact that racial disparities persistent despite accounting for both indidivual and county-level characteristics speaks to the insidious and complex nature of structural racism. Some, like former president of the American Public Health Association, Dr. Camara Jones, have called on us to recognize structural racism as “a system of structuring opportunity and assigning value based on the social interpretation of how one looks;” and the root cause of all differences in any health outcome associated with race.37 As such, racism is an important social determinant of health that necessitates a structural intervention.37 Acknowledging these complex dynamics, several American cities have declared racism as a public health crisis and committed to put racial equity at the core of all city procedures to advocate for policies that improve health in communities of color.38.39 Other cities and counties have made similar declarations, but it is clear that they must be accompanied by novel structural approaches to effectively reduce these disparities, and ultimately end the HIV epidemic.40 

14. Limitations. I would add multiple testing. Are county level variables too broad of an exposure?

We have addressed the concern about multiple testing in response to Reviewer 2 Comment 2. We have included multiple comparisons as a limitation as suggested by the reviewer.

---

## [Decision Letter · Decision Letter 1]

17 Jan 2022

PONE-D-21-13112R1Individual, Community, and Structural Factors Associated with Linkage to HIV Care Among People Diagnosed with HIV in TennesseePLOS ONE

Dear Dr. Ahonkhai,

Thank you for submitting your manuscript to PLOS ONE. After careful consideration, we feel that it has merit but does not fully meet PLOS ONE’s publication criteria as it currently stands. Therefore, we invite you to submit a revised version of the manuscript that addresses the points raised during the review process.

We look forward to receiving your revised manuscript.

Kind regards,

Natalie J. Shook

Academic Editor

PLOS ONE

Journal Requirements:

Reviewers' comments:

Reviewer's Responses to Questions

**Comments to the Author**

1. If the authors have adequately addressed your comments raised in a previous round of review and you feel that this manuscript is now acceptable for publication, you may indicate that here to bypass the “Comments to the Author” section, enter your conflict of interest statement in the “Confidential to Editor” section, and submit your "Accept" recommendation.

Reviewer #1: All comments have been addressed

Reviewer #2: (No Response)

2. Is the manuscript technically sound, and do the data support the conclusions?

Reviewer #1: Yes

Reviewer #2: Yes

3. Has the statistical analysis been performed appropriately and rigorously? 

Reviewer #1: Yes

Reviewer #2: Yes

4. Have the authors made all data underlying the findings in their manuscript fully available?

Reviewer #1: No

Reviewer #2: No

5. Is the manuscript presented in an intelligible fashion and written in standard English?

Reviewer #1: No

Reviewer #2: Yes

6. Review Comments to the Author

Reviewer #1: Minor clarifications/comments:

1. Thank you to the authors for this revised manuscript. I agree with the updated definition of linkage to care, though the text describing the linkage definition could use slight clarification. Can the authors please add more specificity to what is meant by “the first CD4 or HIV-1 RNA test result… after diagnosis”? (line 77-78). I presume you included any CD4/viral load that was collected at least 1 day after the diagnosis – if this is correct, please revise to state “after the date of diagnosis”.

2. I disagree with the statement in the discussion that “increases in teen birth rate were associated with a strong, but statistically significant increase in linkage to HIV care.” I suggest changing strong to “small”, given that the aRR is 1.02 for the association between this teen pregnancy and linkage to care.

Please correct the following omissions/typographical errors and re-run spell check throughout.

1. aRR for poor mental health days in the abstract is not updated with the new analysis – please check this

2. Line 212: I believe emergency room/department should be grouped with inpatient facility (as it is in Table 2). I believe “outpatient facility or emergency room” is an error here – please revise.

3. Line 253: remove inserted text “facilities ranged from 43% to”

4. Line 277: remove comma after “In addition”

5. Line 328: ‘geographic’ is misspelled

Reviewer #2: Revised TN Linkage Study

The authors assessed trends and individual and county-level factors associated with individual linkage to HIV care in TN. The authors did a nice job addressing previous comments. I appreciate the additional clarification in the methods/discussion related to the modelling approach and limitations. I enjoyed reading and applaud the authors for looking at so many social determinants. I have minor comments below.

Abstract/Introduction. No comment.

Methods. No comment.

Results.

1) Minor. How do you interpret the teen birth rate finding? Is this a positive association: increasing teen birth rate is associated with increasing linkage to care? If so, do you think this is the result of confounding? I see this is clarified in the discussion narrative. But may want a sentence in results.

Discussion.

1) Minor. The authors say, in the discussion and abstract, that racial disparities persisted even when adjusting for county-level social determinants of health. However, the modelling approach in the methods (and in Table 2) doesn’t appear to adjust for structural factors, just individual factors… “adjusting for a priori selected individual level covariates in multivariable analysis that were available in the surveillance data and known to be associated with the outcome of interest, including year of and age at diagnosis, sex, race/ethnicity and HIV transmission risk factor.” Does the individual model adjust for county-level social determinants? If so, please add.

2) Minor – line 328, don’t follow the euphemism ‘paints a picture’. I think you’re trying to say that segregation is correlated with geographic factors? Was it specifically correlated with poor mental health and therefore correlated with poor linkage? The point you’re trying to make about segregation is poorly constructed. Suggest revising.

3) Minor. Multiple testing is not specified in the limitations, although the authors suggest it was included. Imagine this was an oversight.

7. PLOS authors have the option to publish the peer review history of their article (what does this mean?). If published, this will include your full peer review and any attached files.

Reviewer #1: No

Reviewer #2: No

---

## [Author Response · Author response to Decision Letter 1]

26 Jan 2022

Reviewer #1: Minor clarifications/comments:

Comment 1. Thank you to the authors for this revised manuscript. I agree with the updated definition of linkage to care, though the text describing the linkage definition could use slight clarification. Can the authors please add more specificity to what is meant by “the first CD4 or HIV-1 RNA test result… after diagnosis”? (line 77-78). I presume you included any CD4/viral load that was collected at least 1 day after the diagnosis – if this is correct, please revise to state “after the date of diagnosis”.

We have updated this text as suggested which now reads as below:

The outcome of interest was linkage to HIV care, which was defined as receipt of the first CD4 or HIV-1 RNA test result captured via TN’s enhanced HIV/AIDS reporting system (eHARS) after the date of diagnosis, and was assessed at 30, 60, and 90 days. 

Comment 2. I disagree with the statement in the discussion that “increases in teen birth rate were associated with a strong, but statistically significant increase in linkage to HIV care.” I suggest changing strong to “small”, given that the aRR is 1.02 for the association between this teen pregnancy and linkage to care.

We agree with this sentiment and have made the suggested change. The updated text reads as below:

Interestingly, increases in teen birth rate were associated with a small, statistically significant increase in linkage to HIV care.

Comment 3: Please correct the following omissions/typographical errors and re-run spell check throughout.

a. aRR for poor mental health days in the abstract is not updated with the new analysis – please check this

b. Line 212: I believe emergency room/department should be grouped with inpatient facility (as it is in Table 2). I believe “outpatient facility or emergency room” is an error here – please revise.

c. Line 253: remove inserted text “facilities ranged from 43% to”

d. Line 277: remove comma after “In addition”

e. Line 328: ‘geographic’ is misspelled

Each of these omissions/typographical errors was addressed. Additional line editing for spelling/grammar was also completed.

Reviewer #2: Revised TN Linkage Study

Comment 1. Minor. How do you interpret the teen birth rate finding? Is this a positive association: increasing teen birth rate is associated with increasing linkage to care? If so, do you think this is the result of confounding? I see this is clarified in the discussion narrative. But may want a sentence in results.

We appreciate this comment. We have not included any interpretation of the findings in the results section, but have included potential confounding to this discussion as an explanation for this finding. The updated text is below:

The reasons for this are not entirely clear, but could reflect intense wrap around services for pregnant and peripartum women to prevent mother-to-child HIV transmission or could reflect potential confounding. 

Comment 2. Minor. The authors say, in the discussion and abstract, that racial disparities persisted even when adjusting for county-level social determinants of health. However, the modelling approach in the methods (and in Table 2) doesn’t appear to adjust for structural factors, just individual factors… “adjusting for a priori selected individual level covariates in multivariable analysis that were available in the surveillance data and known to be associated with the outcome of interest, including year of and age at diagnosis, sex, race/ethnicity and HIV transmission risk factor.” Does the individual model adjust for county-level social determinants? If so, please add.

The individual model adjusts for individual-level determinants, while the county-level model adjusts for both individual and county-level determinants. We have updated the text in the Results section to more clearly present data from the county-level model that supports our assertion about racial disparities. 

Also, in this model which adjusted for both individual and county-level factors, White and Hispanic individuals had an increased risk of 30-day linkage to care compared to Black individuals (aRR 1.33, 95%CI 1.30-1.37, aRR 1.44, 95% CI 1.41-1.47 respectively) [data not shown]. 

We have also updated Figure 2 (below) to reflect data from the county-level model (which adjusted for individual and county-level variables and included a county-by-race/ethnicity interaction term).

We analyzed the marginal probabilities of linkage at 30-days in the four highest-burden metropolitan counties by race/ethnicity and found that Black patients persistently had the lowest probability of 30-day linkage to care as compared to both White and Hispanic individuals when adjusting for individual level factors, and when adjusting for both individual and county-level factors when interacting individual-level race/ethnicity with county of residence [Figure 2]. Racial disparities were least prominent in Davidson County (the county seat of Nashville), which also had the highest marginal probability of linkage to care of the four highest-HIV-burdened counties. 

Comment 3. Minor – line 328, don’t follow the euphemism ‘paints a picture’. I think you’re trying to say that segregation is correlated with geographic factors? Was it specifically correlated with poor mental health and therefore correlated with poor linkage? The point you’re trying to make about segregation is poorly constructed. Suggest revising.

We have updated the language to more clearly explain the likely relationship between racial segregation and county-level geographic factors. The updated text is in the Discussion and reads as below:

Additionally, the high correlation of residential racial segregation (White, non-White) with many county-level factors, while not surprising, further underscores the relationship between race and a range of geographic factors that can impact health.

Comment 4. Minor. Multiple testing is not specified in the limitations, although the authors suggest it was included. Imagine this was an oversight.

We have updated the discussion to include the limitations posed by multiple testing. 

We acknowledge that we have included many covariates in our analyses, and could be subject to limitations from multiple testing.

---

## [Editor Report · Decision Letter 2]

14 Feb 2022

Individual, Community, and Structural Factors Associated with Linkage to HIV Care Among People Diagnosed with HIV in Tennessee

PONE-D-21-13112R2

Dear Dr. Ahonkhai,

We’re pleased to inform you that your manuscript has been judged scientifically suitable for publication and will be formally accepted for publication once it meets all outstanding technical requirements.

Kind regards,

Natalie J. Shook

Academic Editor

PLOS ONE
---

## [Editor Report · Acceptance letter]

18 Feb 2022

PONE-D-21-13112R2 

Individual, Community, and Structural Factors Associated with Linkage to HIV Care Among People Diagnosed with HIV in Tennessee 

Dear Dr. Ahonkhai:

I'm pleased to inform you that your manuscript has been deemed suitable for publication in PLOS ONE. Congratulations! Your manuscript is now with our production department. 

Kind regards, 

on behalf of

Dr. Natalie J. Shook 

Academic Editor

PLOS ONE